# Psoriasis Increases the Risk of Sudden Sensorineural Hearing Loss: A Longitudinal Follow Up Study Using a National Sample Cohort

**DOI:** 10.3390/ijerph17249310

**Published:** 2020-12-12

**Authors:** Hyo Geun Choi, Bumjung Park, Seok Min Hong, Il-Seok Park, Sung Kyun Kim

**Affiliations:** 1Hallym Data Science Laboratory, College of Medicine, Hallym University, Anyang 14068, Korea; pupen@naver.com; 2Department of Otorhinolaryngology-Head & Neck Surgery, Sacred Heart Hospital, Hallym University College of Medicine, Anyang 14068, Korea; bumjung426@gmail.com; 3Department of Otorhinolaryngology-Head & Neck Surgery, Dongtan Sacred Heart Hospital, Hallym University College of Medicine, Dongtan 18450, Korea; thecell20@gmail.com (S.M.H.); ispark@hallym.or.kr (I.-S.P.)

**Keywords:** sudden sensorineural hearing loss, risk, psoriasis, cohort, epidemiology

## Abstract

Psoriasis is a well-known immune-mediated disease. Its autoimmune pathophysiology is consistent with the immune-mediated systemic vascular hypothesis regarding the pathogenesis of sudden sensorineural hearing loss (SSNHL). The purpose of our study was to investigate whether psoriasis affects the prevalence of SSNHL in all age groups matched by age, sex, income, and region of residence. Korean Health Insurance Review and Assessment Service-National Patient Samples were collected from 2002 to 2013. A 1:4 matched psoriasis group (n = 12,864) and control group (n = 51,456) were selected. The crude (simple) and adjusted (Charlson comorbidity index) hazard ratios (HR) for psoriasis and SSNHL were analyzed using the stratified Cox proportional hazard model. The incidence of SSNHL was significantly higher in the psoriasis group than in the control group (0.5% vs. 0.4%, *p* = 0.004). Psoriasis increased the risk of SSNHL (adjusted HR = 1.44, 95% confidence interval (CI) = 1.09–1.90, *p* = 0.010). In the stratification analysis, the incidence of SSNHL was significantly higher in the 30–59-year-old group than other group SSNHL (adjusted HR = 1.50, 95% CI = 1.06–2.12, *p* = 0.023). In addition, SSNHL occurred significantly more frequently in men with psoriasis (adjusted HR = 1.70, 95% CI = 1.17–2.49, *p* = 0.006). Psoriasis increased the risk of SSNHL, and SSNHL was more prevalent in between the age of 30–59-year-olds and men with psoriasis.

## 1. Introduction

Sudden sensorineural hearing loss (SSNHL) is defined as a minimum 30 dB hearing loss over three consecutive frequencies on pure tone audiometry, occurring within 72 h [1]. Its annual incidence is estimated at 5–20/100,000 people [1]. Various factors, such as viral infection and metabolism, autoimmune, and central nervous system pathology, have been reported as the cause of SSNHL, but most cases of SSNHL are considered idiopathic [2]. Immunologically, it is believed that SSNHL is caused by the interaction between circulating antibodies and an antigen in the inner ear, or by direct inner ear by activated T cells [3]. In addition, activation of the complement system can be a mechanism of inner ear damage [4]. Immuno-suppressive therapy and corticosteroids have been shown to be effective treatment options for SSNHL, which may be associated with immune-mediated pathophysiology. As with immune-mediated sensorineural hearing loss, SSNHL with immune-mediated etiology is difficult to diagnose because of the lack of tests and biomarkers that can define the immunologic cause [4,5]. However, there are reports that such autoantibodies as anti-endothelial antibodies, anticardiolipin, anti-beta2-glycoprotein-1, and anti-heat shock protein-70 antibodies are involved in SSNHL [6,7], and the development of SSNHL in patients with systemic lupus erythematosus (SLE), psoriasis, and Wegener granulomatosis supports the immune-mediated etiology of SSNHL [8,9,10].

Psoriasis is an immune-mediated, chronic inflammatory disease that manifests as both skin and systemic symptoms [11]. T helper (Th)-1 and -17 cell activations are typical immunological mechanisms that cause systemic inflammation [12]. This leads to the formation of proinflammatory mediators, which results in the dysfunction of endothelial cells and platelet activation [13]. Cardiovascular disease, diabetes mellitus, and malignancy are more common in patients with psoriasis, as the symptoms of systemic manifestations are related to various diseases [14,15,16]. Psoriasis is likely to be associated with inflammatory mediators that are produced when systemic inflammatory processes occur. A few studies have reported that psoriasis can also be associated with sudden hearing loss since both diseases may share an immune-mediated and systemic inflammatory pathogenesis [9,17].

Psoriasis and SSNHL are frequently encountered in dermatology and otorhinolaryngology, but their association has been rarely reported as a case report [18], despite the pathophysiological hypothesis. The present cohort study aimed to investigate the association of psoriasis on the prevalence of SSNHL using a large-scale, nationwide, population-based cohort sample.

## 2. Materials and Methods

### 2.1. Study Population and Data Collection

This national cohort study relied on data from the Korean Health Insurance Review and Assessment Service-National Sample Cohort (HIRA-NSC). The detailed description of these data was described in our previous studies [19,20].

### 2.2. Participant Selection

Out of 1,125,691 cases with 114,369,638 medical claim codes, we included participants who were diagnosed with psoriasis (ICD-10: L40). From them, we selected participants who treated had undergone treatment for psoriasis ≥2 times (n = 12,921). Therefore, the participants were followed for up to 12 years.

SSNHL was diagnosed using ICD-10 codes (H912). We only included participants who have undergone an audiometry exam (claim code: E6931-E6937, F6341-F6348) and who were treated with steroids. From 2002 to 2013, 5244 SSNHL patients were selected following our previous studies [21,22].

The psoriasis participants were matched 1:4 with participants (control group) who had never been diagnosed with psoriasis between 2002 and 2013. The control groups were selected from the base population (n = 1,112,770). The patients and controls were matched for age, group, sex, income group, and region of residence. To prevent selection bias when selecting the matched participants, the control group participants were sorted using a random number order and selected from top to bottom. Index date was determined as the diagnosis date of psoriasis. We assumed that the matched control participants were involved at the same time as the psoriasis participants to whom they were matched (index date). Therefore, any patient who had died before the index date was excluded from the control group. In both the psoriasis and control groups, individuals who had a history of SSNHL before the index date were excluded. In the psoriasis group, 25 participants were excluded. Psoriasis group members for whom we could not identify enough matching participants were excluded (n = 32). Finally, 1:4 matching resulted in the inclusion of 12,864 participants in the psoriasis and 51,456 in the control group (Figure 1).

### 2.3. Variables

The variables of age group, sex, income, and region of residence were determined following our previous studies [23,24]. Age groups were classified as 0–4, 5–9, 10–14…, and 85+ years old. Charlson comorbidity index (CCI) was used for 17 comorbidities as the continuous variable (0 (no comorbidity) through 29 (multiple comorbidities)) [25].

### 2.4. Statistical Analyses

The chi-square test was used to compare the rates of general characteristics between the psoriasis and control groups.

To analyze the hazard ratio of psoriasis on SSNHL, the stratified Cox proportional hazard model was used. In this analysis, crude (simple) and adjusted (CCI) models were used. In this analysis, we stratified age, sex, income, and region of residence.

For the subgroup analysis, we divided the participants by age and sex (0–29 years, 30–59 years, 60+ years; male and female).

Two-tailed analyses were conducted, and P values less than 0.05 were considered significant. The results were statistically analyzed using SPSS v. 21.0 (IBM, Armonk, NY, USA). 

### 2.5. Availability of Data and Materials

Releasing of the data by the researcher is not allowed legally. All of data are available from the database of National health Insurance Sharing Service (NHISS) https://nhiss.nhis.or.kr/ NHISS allows all this data for the any researcher who promises to follow the research ethics with some cost. If you want to access the data of this article, you could download it from the website after promising to follow the research ethics.

## 3. Results

The distributions of age, sex, income level, and region of residence were comparable between the psoriasis and control groups (Table 1). The study included 12,864 patients in the psoriasis group and 51,456 matched participants in the control group, four times the number in the psoriasis group. The sex ratio for the psoriasis group was 55.4% (n = 7129) males and 44.6% (n = 5735) females; the same ratio was established for the control group. The proportion of patients in the psoriasis group that lived in urban areas was 47.8% (n = 6146), while 52.2% (n = 6718) lived in rural areas. The rate of SSNHL was 0.54% (n = 69) in the psoriasis group and 0.36% (n = 185) in the control group, which was significantly higher in the psoriasis group (p = 0.004). The crude and adjusted hazard ratios (HR) were 1.49 (95% CI = 1.13–1.97, *p* = 0.005) and 1.44 (95% CI = 1.09–1.90, *p* = 0.010), respectively (Table 2). A subgroup analysis found that the adjusted HR of SSNHL was high in the 30–59-year-olds compared with that in the control group (adjusted HR = 1.44, 95% CI = 1.06–2.12, *p* = 0.023) (Table 3). In addition, the risk of SSNHL of men with psoriasis was elevated even after adjusting for confounders (HR = 1.70, 95% CI = 1.17–2.49, *p* = 0.006) (Table 3).

## 4. Discussion

Psoriasis increased the risk of SSNHL in this nationwide cohort with a 12-year follow-up from 2002 to 2013. Psoriasis significantly increased the risk of SSNHL among middle-aged (30–59 years) and male group in this national sample cohort when age, sex, income, and region were matched.

In a similar study using national cohort data in Taiwan, the incidence of SSNHL was significantly higher in the psoriasis cohort than in the control cohort [9]. Additionally, the incidence of SSNHL was significantly higher among psoriasis patients aged 35 to 49 years and those older than 65 years [9]. Although the data for Taiwan had a slightly different age classification than this cohort, the studies were similar in that both found an incidence of SSNHL that was significantly higher in the middle-age group (30–59 years) and the old-age group (over 60 years). Yen et al. reported a significant difference in the incidence of SSNHL in all genders of the psoriasis cohort compared with the control cohort [9]. However, in this study, we analyzed the participants by grouping them in terms of age and gender. The incidence of SSNHL in females with psoriasis was not higher than in control cohort for all ages; it was significantly higher only in the abovementioned age groups (30–59 years) and males with psoriasis. Although autoimmune diseases such as psoriasis are known to occur in women [26], it is noteworthy that in this study the incidence of SSNHL was higher in men over the age of 30 years. In addition, although it is difficult to accurately determine the reason for such a result, factors such as the difference in population composition by age group, gender ratio, and rate of medical use by age in South Korea may have affected it. In the next study, it is necessary to investigate pathophysiological association using these epidemiological results.

One immunological cause of SSNHL is impairment of the micro-vascular supply to cochlea, which is known to be caused by autoantibodies that promote thrombotic conditions in the labyrinthine vessels [27]. In psoriasis, systemic vascular inflammation is caused by interleukin (IL) -6, IL-12, IL-23, tumor necrosis factor (TNF) -α production and oxidative stress, which induce thrombosis in the vessels and lipid dysregulation [28,29]. These two diseases share immunological aspects but there is little research into the relationship between immunological factors and the development of diseases. Moreover, studies of the association of autoimmune disease with SSNHL in a nationwide cohort are extremely rare. Lin et al. analyzed the incidence of SSNHL in SLE patients using nation-wide cohort data [8]. They found that SLE which, like psoriasis, is an autoimmune disease was also significantly associated with an increased incidence SSNHL and that the incidence of SSNHL was significantly higher in young people (0–34 years) than in middle-aged and older people [8]. In addition, the incidence of SSNHL was significantly higher in women than in the control group, and the onset of SSNHL was higher 1–3 years and 3–5 years after the diagnosis of SLE compared with less than one year and more than five years after diagnosis [8]. Cross-sectional data from the UK biobank show that Meniere’s disease affects the incidence of various autoimmune diseases including psoriasis [30]. Frejo et al. were demonstrated that locus PSORC1 and the allelic variant rs4947296, both located 6p21.33 mediate inflammation of the blood-labyrinth barrier and endolymphatic sac by regulation of gene expression and upregulation of NF-κB translation in the TNF-like weak inducer of apoptosis/Fn14 pathway in bilateral Meniere’s disease [31].

In psoriasis, proinflammatory cytokines such as tumor necrosis factor-alpha (TNF-α) and interleukin (IL)-6 induce endothelial dysfunction, thereby increasing the risk of cardiovascular disease and increasing the incidence of cerebral ischemic disease [32,33]. In addition, these cytokines can induce central obesity and insulin resistance by stimulating the hypothalamic-pituitary axis [33]. Recent studies have shown that psoriasis is associated with migraine, as the endothelial dysfunction caused by increased TNF leads to vascular impairment [34,35]. Chu et al. reported that migraine is a risk factor for SSNHL in a nationwide cohort, which is a result that can show that proinflammatory cytokines can be linked to the potential relation between psoriasis and SSNHL [36]. Additionally, psoriasis has been associated with chronic rhinosinusitis (CRS) without nasal polyps (NP) due to Th 1 immune dysregulation [37], and the hazard ratio of CRS without NP in psoriasis patients was 2.01 (95% CI = 1.54–2.62) in a population-based cohort study [38].

Since SSNHL is potentially an immune-mediated disease, several studies have attempted to find an association between it and a well-established autoimmune disease. However, because it is rare for both diseases to occur at the same time or within a short period of time, researchers face the difficulty of tracking patients for long periods, which may be a limitation of these kinds of studies. Recently, microRNA, gene mutation, and polymorphism analysis have been investigated to be associated with hearing loss in psoriasis. Nine micro RNAs related to hearing loss and 12 target genes were identified by bioinformatic analysis from plasma samples of psoriasis patients [39]. Moreover, mutations in the connexin26 (Cx26)-encoding gap junction beta-2 gene expressed in cells of the organ of Corti and downregulation of Cx26 were seen in psoriatic patients [39]. Down-regulation of Cx26 promotes cell death of hair cells and stria vascularis in cochlea, impairs gap junctions of skin cells, and can be associated with various psoriatic skin manifestations [40,41].

This study has several advantages compared with other studies. The data used in the analysis came from a nationwide cohort, and the most important advantage is that the database was constructed by matching patients and controls in terms of age, sex, residence area, and income factors. Controlling the many confounding factors in the analysis of the incidence of SSNHL as a dependent variable may have increased the reliability of the results. An additional advantage is that the inclusion criteria for SSNHL patients were clear. Although nationwide cohort studies have the advantage of being able to include a large number of subjects, recall bias may occur if the variable being analyzed is a symptom that is not a disease entity; furthermore, it is difficult to establish inclusion criteria and to obtain reliable results. In the present study, evidence of SSNHL was obtained from the patient’s medical records and the HIRA data system. We supported the diagnostic evidence by using pure tone audiometry test results, prescriptions, and medical claim codes as inclusion criteria, which prevented recall bias. Because HIRA data are kept for all people in the nation, the subjects included in the analysis were not missed during follow-up. In addition, because the time of diagnosis was clear, the confounding effect caused by the history of SSNHL before the onset of psoriasis was minimized.

Although we found that psoriasis increased the incidence of SSNHL, there are a few weak points in our research. We could not determine the degree of hearing impairment at the time of diagnosis because we did not obtain hearing thresholds on audiograms for all subjects. In addition, general health data for of individual patients, such as height, weight, smoking, drinking, and the history of noise exposure, are difficult to identify in the HIRA data system. There are various causes of SSNHL such as infection, head trauma, autoimmune diseases, drugs, blood circulation, and neurologic disorders. However, descriptive information that can be clearly identified and sub-grouped according to factors cannot be obtained in the data provided to researchers. Specific information regarding skin lesions, autoantibody titers, and systemic manifestations in patients with psoriasis are also unavailable. Thus, future studies will need to determine whether SSNHL is associated with the factors that represent the disease activity of psoriasis and which causative factors of SSNHL are associated with psoriasis. In addition, all studies on association between SSNHL and psoriasis through nationwide, population-based studies have been conducted in Asian populations, and no epidemiological studies are available in European descendant population. Based on the results of this study, it will be possible to establish evidence for the association between the two diseases by expanding it to a nationwide cohort targeting various races and countries.

The results suggest that doctors who are responsible for patients with psoriasis need to be aware of hearing impairment during the follow-up period. In addition, because SSNHL is an otologic emergency, early detection is important. Therefore, patients with psoriasis should be informed of the possibility of developing sudden hearing impairment and should be aware of appropriate early treatment.

## 5. Conclusions

Psoriasis was significantly associated with an increased incidence of SSNHL. Moreover, middle-aged (30–59 years old) psoriasis patients, particularly males, have a significantly higher risk of developing SSNHL compared with younger patients. Therefore, it is necessary to pay attention to hearing impairments in male, middle aged psoriasis patients and to inform to psoriasis patients about the possibility of developing sudden hearing loss.

## Figures and Tables

**Figure 1 ijerph-17-09310-f001:**
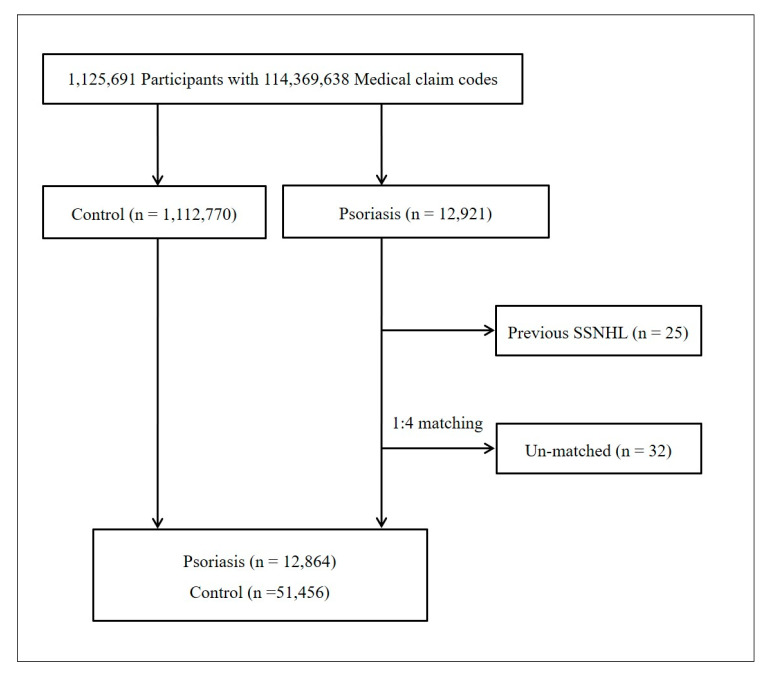
A schematic illustration of the participant selection process used in the present study. Of 1,125,691 participants, 12,864 psoriasis participants were matched with 51,456 control participants for age, group, sex, income group, region of residence, and past medical histories.

**Table 1 ijerph-17-09310-t001:** General characteristics of participants.

Characteristics	Total Participants		
	Psoriasis (n, %)	Control (n, %)	*p*-Value
Age (years old)			1.000
0–4	127 (1.0)	508 (1.0)	
5–9	251 (2.0)	1004 (2.0)	
10–14	460 (3.6)	1840 (3.6)	
15–19	614 (4.8)	2456 (4.8)	
20–24	785 (6.1)	3140 (6.1)	
25–29	995 (7.7)	3980 (7.7)	
30–34	1138 (8.8)	4552 (8.8)	
35–39	1207 (9.4)	4828 (9.4)	
40–44	1235 (9.6)	4940 (9.6)	
45–49	1264 (9.8)	5056 (9.8)	
50–54	1179 (9.2)	4716 (9.2)	
55–59	937 (7.3)	3748 (7.3)	
60–64	807 (6.3)	3228 (6.3)	
65–69	715 (5.6)	2860 (5.6)	
70–74	583 (4.5)	2332 (4.5)	
75–79	317 (2.5)	1268 (2.5)	
80–84	167 (1.3)	668 (1.3)	
85+	83 (0.6)	332 (0.6)	
Sex			1.000
Male	7129 (55.4)	28,516 (55.4)	
Female	5735 (44.6)	22,940 (44.6)	
Income			1.000
1 (lowest)	258 (2.0)	1032 (2.0)	
2	844 (6.6)	3376 (6.6)	
3	828 (6.4)	3312 (6.4)	
4	937 (7.3)	3748 (7.3)	
5	1066 (8.3)	4264 (8.3)	
6	1118 (8.7)	4472 (8.7)	
7	1294 (10.1)	5176 (10.1)	
8	1337 (10.4)	5348 (10.4)	
9	1540 (12.0)	6160 (12.0)	
10	1734 (13.5)	6936 (13.5)	
11 (highest)	1908 (14.8)	7632 (14.8)	
Charlson Comorbidity index			<0.001 *
0	5674 (44.1)	25,311 (49.2)	
1	1553 (12.1)	5990 (11.6)	
2	1518 (11.8)	5721 (11.1)	
3	1231 (9.6)	4373 (8.5)	
≥4	2888 (22.5)	10,061 (19.6)	
SSNHL	69 (0.5)	185 (0.4)	0.004 *

SSNHL: sudden sensorineural hearing loss; * Chi-square test. Significance at *p* < 0.05.

**Table 2 ijerph-17-09310-t002:** Crude and adjusted hazard ratios (95% confidence interval) of psoriasis for sudden sensorineural hearing loss.

Characteristics		Hazard Ratio (95% CI)			
		Crude	*p*-Value	Adjusted ^†,‡^	*p*-Value
Psoriasis			0.005 *		0.010 *
	Yes	1.49 (1.13–1.97)		1.44 (1.09–1.90)	
	No	1.00		1.00	

* Cox-proportional hazard regression model, significance at *p* < 0.05; ^†^ stratified model for age, sex, income, and region of residence; ^‡^ adjusted model for Charlson comorbidity index.

**Table 3 ijerph-17-09310-t003:** Subgroup analysis of crude and adjusted hazard ratios (95% confidence interval) of psoriasis for sudden sensorineural hearing loss according to age and sex.

Chaacteristics		Hazard Ratio (95% CI)			
		Crude	*p*-Value	Adjusted ^†,‡^	*p*-Value
0–29 years old, n = 16,160					
Psoriasis			0.381		0.471
	Yes	1.47 (0.62–3.50)		1.38 (0.57–3.29)	
	No	1.00		1.00	
30–59 years old, n = 34,800					
Psoriasis			0.011 *		0.023 *
	Yes	1.57 (1.11–2.21)		1.50 (1.06–2.12)	
	No	1.00		1.00	
60+ years old, n = 13,360					
Psoriasis			0.311		0.329
	Yes	1.33 (0.77–2.30)		1.32 (0.76–2.28)	
	No	1.00		1.00	
Men, n = 35,645					
Psoriasis			0.003 *		0.006 *
	Yes	1.77 (1.22–2.58)		1.70 (1.17–2.49)	
	No	1.00		1.00	
Women, n = 28,675					
Psoriasis			0.313		0.390
	Yes	1.24 (0.82–1.86)		1.20 (0.79–1.81)	
	No	1.00		1.00	

* Cox-proportional hazard regression model, significance at *p* < 0.05; ^†^ stratified model for age, sex, income, and region of residence; ^‡^ adjusted model for Charlson comorbidity index.

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
