# Peer review of "Psoriasis Increases the Risk of Sudden Sensorineural Hearing Loss: A Longitudinal Follow Up Study Using a National Sample Cohort"

_ijerph, 2020, doi:10.3390/ijerph17249310_

Round 1
Reviewer 1 Report
This is a well-design cross-sectional study to investigate the association of SSNHL with psoriasis. I would recommend its publication after some minor changes.
Introduction
This is fine, but the aim of the study should be reformulated: “to investigate the effect of psoriasis on the prevalence of SSNHL using a large-scale, nationwide, population-based cohort…” The cross-sectional design only can demonstrate association, no causality.
Please, revise as “to investigate the association of psoriasis with SSNHL using a large…
Methods
Participant selection of individuals with SSNHL includes hearing test and response to steroids. The design 1:4 included 12864 participants in the psoriasis group and 51456 in the control group.
Results
In Table 3, Psoriasis and SNSHL are not associated. Could the authors explain the reason for this?
Discussion
I will suggest the moderation in some statements.
“Psoriasis significantly increased the risk of SSNHL among middle-aged male group 138 (30-59 years) and older males (over 60 years) in this national sample cohort when age, sex, income, and region were matched.”
This is not correct, according to data presented in Table 3, in men >60 years old there is no association. Please, revise this sentence and limit the discussion to your findings. The extension to older people does not help.
Psoriasis has been associated with migraine. Please comment this in the discussion and the potential relation with SSNHL.
Psoriasis has also been associated with Meniere disease in UKBiobank data. The locus PSORC1 and the allelic variant rs4947296, both located close in chromosome 6 could explain this association. Please comment in the discussion.
In the discussion, you should indicate that the 2 studies published showing association between SSNHL and psoriasis have been conducted in Asian population and no epidemiological studies are available in European descendant population.
Suggested references.
- Frejo L, Requena T, Okawa S, et al. Regulation of Fn14 Receptor and NF-κB Underlies Inflammation in Meniere’s Disease. Front Immunol. 2017;8(DEC):1739. doi:10.3389/fimmu.2017.01739
- Tyrrell JS, Whinney DJD, Ukoumunne OC, Fleming LE, Osborne NJ. Prevalence, associated factors, and comorbid conditions for Ménière’s disease. Ear Hear. 2014;35(4):e162–9.
- Chu CH, Liu CJ, Fuh JL, Shiao AS, Chen TJ, Wang SJ. Migraine is a risk factor for sudden sensorineural hearing loss: A nationwide population-based study. Cephalalgia. 2012;33(2):80-86.
Author Response
The revisions made after carefully considering the comments of the reviewers and editor are as follows. (Note: reviewer comments are in italics; our responses are in red.)
Reviewer #1:
This is a well-design cross-sectional study to investigate the association of SSNHL with psoriasis. I would recommend its publication after some minor changes.
Introduction
This is fine, but the aim of the study should be reformulated: “to investigate the effect of psoriasis on the prevalence of SSNHL using a large-scale, nationwide, population-based cohort…” The cross-sectional design only can demonstrate association, no causality.
Please, revise as “to investigate the association of psoriasis with SSNHL using a large…
: Thank you for your insightful comments. The ‘effect’ in the last sentence of the introduction section that you pointed out was modified to ‘association’. The correction of the word has made it possible to better highlight the purpose of our study.
Methods
Participant selection of individuals with SSNHL includes hearing test and response to steroids. The design 1:4 included 12864 participants in the psoriasis group and 51456 in the control group.
Results
In Table 3, Psoriasis and SNSHL are not associated. Could the authors explain the reason for this?
: Thank you for your valuable comment. The authors already describe for subgroup analysis in Table 3 from lines 121 to 124 and from line 148-152. In the subgroup analysis according to age and sex, the SSNHL incidence of the psoriasis group was significantly higher in the 30-59-year-old group (adjusted HR = 1.44, 95% CI = 1.06-2.12, p=0.023) and the men group than in the control group (adjusted HR = 1.70, 95% CI = 1.17-2.49, p=0.006). Although it is difficult to accurately determine the reason for such a result, factors such as the difference in population composition by age group, gender ratio, and rate of medical use by age in South Korea may have affected it. In the next study, it is necessary to investigate pathophysiological association using these epidemiological results. We added a paragraph like this to lines 152 to 155 in discussion section. Your critical review has allowed us to become a more complete manuscript.
Discussion
I will suggest the moderation in some statements.
“Psoriasis significantly increased the risk of SSNHL among middle-aged male group 138 (30-59 years) and older males (over 60 years) in this national sample cohort when age, sex, income, and region were matched.”
This is not correct, according to data presented in Table 3, in men >60 years old there is no association. Please, revise this sentence and limit the discussion to your findings. The extension to older people does not help.
: Thank you for the exact point-out. As you mentioned, the SSNHL incidence of the psoriasis group in the old age group over 60 was not significantly different from that of the control group. Therefore, the first paragraph of the discussion section has been modified to match the results.
Psoriasis has been associated with migraine. Please comment this in the discussion and the potential relation with SSNHL.
: We are grateful to the reviewer for catching this issue. We described in lines 180-182 of the discussion section that there was association between psoriasis and migraine in a nationwide population-based cohort by referring to the reference you suggested. Your comment has allowed this paragraph to scientifically support that both diseases have a pathophysiological relationship with proinflammatory cytokines.
Psoriasis has also been associated with Meniere disease in UKBiobank data. The locus PSORC1 and the allelic variant rs4947296, both located close in chromosome 6 could explain this association. Please comment in the discussion.
: Thank you for your valuable suggestions. We described the association of psoriasis and Meniere's disease using your recommended reference. in lines 169-174 of the discussion section. We are grateful for allowing us to add that Meniere's disease, which has a relatively high incidence in the otology field, is associated with the autoimmune mediated inflammatory response in addition to SSNHL.
In the discussion, you should indicate that the 2 studies published showing association between SSNHL and psoriasis have been conducted in Asian population and no epidemiological studies are available in European descendant population.
: Thank you so much for your valuable comment. We described in lines 223-227 of the discussion section that a nationwide, population-based study of the association between SSNHL and psoriasis was conducted only in the Asian population and needs to be studied in various races and countries in the future.
Suggested references.
- Frejo L, Requena T, Okawa S, et al. Regulation of Fn14 Receptor and NF-κB Underlies Inflammation in Meniere’s Disease. Front Immunol. 2017;8(DEC):1739. doi:10.3389/fimmu.2017.01739
- Tyrrell JS, Whinney DJD, Ukoumunne OC, Fleming LE, Osborne NJ. Prevalence, associated factors, and comorbid conditions for Ménière’s disease. Ear Hear. 2014;35(4):e162–9.
- Chu CH, Liu CJ, Fuh JL, Shiao AS, Chen TJ, Wang SJ. Migraine is a risk factor for sudden sensorineural hearing loss: A nationwide population-based study. Cephalalgia. 2012;33(2):80-86.
Reviewer 2 Report
This is a well-written article and can provide further information in the scientific literature on the risk of SSNHL in patients with psoriasis. The following revisions are suggested:
Methodology
The methodology in the study is flawed in that it fails to identify other factors in the participants which may have led to SSNHL instead of the psoriasis. See list of factors cited by the National Institute on Deafness and Other Communication Disorders (NIDCD): https://www.nidcd.nih.gov/health/sudden-deafness#2
Infections
Head trauma
Autoimmune diseases
Exposure to certain drugs that treat cancer or severe infections
Blood circulation problems
Neurological disorders, such as multiple sclerosis
Disorders of the inner ear, such as Ménière’s disease
In addition, there is no mention on whether medications that participants were taking to treat the psoriasis could been ototoxic and contributed to the SSHL. See: Mahasitthiwat, V. (2005). A woman with sudden bilateral sensorineural hearing loss after treatment psoriasis with acitretin. Journal-Medical Association of Thailand, 88, S79.
At best these data should be re-analyzed including the factors above, or at minimum, the absence should be mentioned as a study limitation.
Literature Review
There is only one reference directly linking the topic of hearing loss with psoriasis, though there have been several published studies in this area. The authors need to do a more thorough review and report of the literature in this area.
Author Response
The revisions made after carefully considering the comments of the reviewers and editor are as follows. (Note: reviewer comments are in italics; our responses are in red.)
Reviewer #2
This is a well-written article and can provide further information in the scientific literature on the risk of SSNHL in patients with psoriasis. The following revisions are suggested:
Methodology
The methodology in the study is flawed in that it fails to identify other factors in the participants which may have led to SSNHL instead of the psoriasis. See list of factors cited by the National Institute on Deafness and Other Communication Disorders (NIDCD): https://www.nidcd.nih.gov/health/sudden-deafness#2
Infections
Head trauma
Autoimmune diseases
Exposure to certain drugs that treat cancer or severe infections
Blood circulation problems
Neurological disorders, such as multiple sclerosis
Disorders of the inner ear, such as Ménière’s disease
In addition, there is no mention on whether medications that participants were taking to treat the psoriasis could been ototoxic and contributed to the SSHL. See: Mahasitthiwat, V. (2005). A woman with sudden bilateral sensorineural hearing loss after treatment psoriasis with acitretin. Journal-Medical Association of Thailand, 88, S79.
At best these data should be re-analyzed including the factors above, or at minimum, the absence should be mentioned as a study limitation.
: Thank you for your insightful comments. As you mentioned, there are several factors that may be responsible for SSNHL, such as infection, head trauma, autoimmune diseases, drugs, blood circulation, and neurologic disorders. The causative factor of SSNHL is a very important part of hearing improvement. Therefore, analyzing the risk by subdividing the group according to the causative factor would have led to more reliable results in this study.
However, since sample cohorts in this study based on medical claim codes such as examination, prescription of medication, and ICD-10 diagnostic code, it is very complex to analyze the results according to the cause of the disease in terms of coding, number of participants by group, and statistical analysis (line 216-223). We describe these limitations in detail in the discussion section. Also, reference was added to the last paragraph of introduction section for the case report of bilateral SSNHL related to psoriasis treatment (acitretin).
Literature Review
There is only one reference directly linking the topic of hearing loss with psoriasis, though there have been several published studies in this area. The authors need to do a more thorough review and report of the literature in this area.
: Thank you for your valuable comment. As we mentioned in the discussion, there are few epidemiological studies on the association between psoriasis and SSNHL.
Recent published studies on microRNA, gene mutation, and polymorphism between psoriasis and hearing loss are described in the discussion section (line 190-196). The fact that factors such as GJB2 gene mutation and down-regulation of Cx26 can cause skin manifestation and hearing loss can further support the epidemiological results of our study.